# GatePro: Parameter-Free Expert Selection Optimization for Mixture-of-Experts Models

## Abstract

Modern large language models leverage Mixture-of-Experts (MoE) architectures for efficient scaling, but face a critical challenge: functionally similar experts are often selected simultaneously, creating redundant computation and limiting effective model capacity. Existing auxiliary balance loss methods improve token distribution but fail to address the underlying expert diversity problem. We introduce GatePro, a novel parameter-free method that directly promotes expert selection diversity. GatePro identifies the most similar expert pairs and introduces localized competition mechanisms, preventing redundant expert co-activation while maintaining natural expert specialization. Our comprehensive evaluation demonstrates GatePro's effectiveness across model scales and benchmarks. Analysis demonstrates GatePro's ability to achieve enhanced expert diversity, where experts develop more distinct and complementary capabilities, avoiding functional redundancy. This approach can be deployed hot-swappable during any training phase without additional learnable parameters, offering a practical solution for improving MoE effectiveness.

## 1 Introduction

The rapid advancement of large language models (LLMs) has demonstrated remarkable capabilities across diverse natural language processing tasks (Achiam et al., 2023; Touvron et al., 2023; Seed et al., 2025). However, scaling dense models faces significant challenges, including rising computational costs, memory requirements, and training instabilities (Kaplan et al., 2020; Hoffmann et al., 2022). As model parameters grow from billions to trillions, the computational burden becomes prohibitive for both training and inference. Meanwhile, performance gains often plateau or become marginal (Wei et al., 2022; Tay et al., 2022).

Mixture of Experts (MoE) models have emerged as a compelling solution to address these scalability challenges by conditionally activating only a subset of parameters for each input token (Shazeer et al., 2017; Fedus et al., 2022). In transformer architectures, MoE layers replace the traditional feed-forward network (FFN) with multiple parallel experts, each a specialized FFN (Lepikhin et al., 2020). A gating mechanism routes each token to its top-$k$ experts, while others remain inactive, enabling dramatic scaling of model capacity while maintaining manageable computational costs (Artetxe et al., 2021; Du et al., 2022). However, early MoE implementations suffer from severe load imbalancing issues, where a few experts receive the majority of tokens while others are underutilized (Shazeer et al., 2017; Bengio et al., 2013). To address this challenge, auxiliary balance loss functions were introduced (Shazeer et al., 2017; Fedus et al., 2022; Zhou et al., 2022; Lepikhin et al., 2020). These methods successfully improve expert load distribution by penalizing uneven token assignments, ensuring that computational resources are better utilized across all experts.

While auxiliary balance loss effectively addresses load balancing, it overlooks a more fundamental problem: **expert selection diversity**. We observe that MoE models experience significant expert activation delays during early pre-training, where the gating mechanism initially concentrates tokens on a few dominant experts and only gradually expands to utilize additional experts as training progresses. This narrow initial focus creates a cascade of problems: many experts remain undertrained during crucial early learning phases, limiting the model's ability to leverage its full capacity during foundational learning. Moreover, even after the gating mechanism eventually learns to distribute

tokens more broadly, similar experts tend to be co-activated, creating functional redundancy rather than true specialization.

The fundamental issue stems from current expert selection mechanisms focusing solely on load balance while ignoring functional diversity. Auxiliary balance losses (Zhou et al., 2022; Fedus et al., 2022) achieve uniform token distribution by encouraging broader expert utilization over time. However, they operate independently of expert functionality. This approach fails to address the core problem that functionally similar experts can still be co-activated as long as load balance is maintained. Consequently, even when all experts receive adequate tokens, the selected subset for any given input may exhibit significant functional overlap, compromising representational capacity, particularly in deeper layers where expert specialization is crucial for optimal performance.

To address the expert selection diversity challenge, we propose GatePro, a novel parameter-free approach that directly promotes diverse expert selection through localized competition mechanisms. Our motivation stems from the observation that expert selection can be viewed as a competitive propagation process between experts, where the influence of one expert's selection propagates to affect others based on their functional similarities. GatePro employs targeted localized competition between the most similar expert pairs, ensuring that functionally redundant experts cannot be simultaneously selected while preserving natural specialization for dissimilar experts. This competitive propagation enables GatePro to achieve enhanced expert diversity, where experts develop more distinct and complementary capabilities, avoiding functional redundancy.

To validate GatePro's effectiveness, we conduct extensive experiments across different model scales, varying expert pool sizes, and multiple training stages. We evaluate GatePro during both pretraining from scratch and continued training (CT) phases, tracking performance from early to advanced stages to demonstrate robustness across different training scenarios. Additionally, we perform analysis to understand how GatePro achieves improved expert utilization and selection diversity.

In summary, this work makes the following contributions:

(1). We identify expert selection diversity as a fundamental challenge overlooked by existing MoE approaches and propose GatePro, a parameter-free approach that promotes diverse expert selection through competitive propagation between functionally similar experts. GatePro is hot-swappable, allowing expert diversity enhancement without additional learnable parameters.

(2). We provide comprehensive experimental evaluation across multiple model scales and benchmarks, demonstrating that GatePro consistently outperforms baseline MoE models, with particularly strong benefits during all training phases.

(3). We conduct comprehensive mechanistic analysis through expert utilization tracking and gating similarity evaluation, revealing that GatePro significantly accelerates expert activation, reduces expert similarity, increases selection entropy, and maintains diversity patterns, with particularly significant improvements in deeper layers where expert specialization is most critical.

## 2 RELATED WORK

**Sparse MoE and routing.** MoE architectures scale model capacity through sparse activation and learned routing. As a foundational contribution, Shazeer et al. (2017) introduced sparsely-gated layers with token-choice routing. Building on this, Switch Transformer (Fedus et al., 2022) simplified the original design and proposed the widely used load-balancing loss (LBL) to encourage balanced expert utilization in large language models. ST-MoE (Zoph et al., 2022) identified instability caused by LBL and introduced the z-loss, which regularizes router logits to maintain stable magnitude. GShard (Lepikhin et al., 2020) further combined auxiliary balancing with expert capacity limits and a random routing strategy for secondary expert selection. To mitigate persistent imbalance, Lewis et al. (2021) proposed the BASE layer, reframing token–expert assignment as a linear assignment problem, while Clark et al. (2022) extended this with optimal transport formulations and a reinforcement learning-based router. Other work focuses on smoothing Top-$k$ routing decisions: DSelect-k (Hazimeh et al., 2021) approximates Top-$k$ with a differentiable formulation, while Dong et al. (2025) cast routing as a minimum-cost maximum-flow problem with a differentiable SoftTopK operator to improve efficiency and balance. Skywork-MoE (Wei et al., 2024) introduced gating-logit normalization and layerwise adaptive coefficients to stabilize training and

encourage diversity, while AdaMoE (Zeng et al., 2024) dynamically adjusts the number of experts per token. Our proposed GatePro differs from these approaches: rather than auxiliary losses or heavy optimization, it employs a loss-free gating mechanism with localized competition between similar gates, simultaneously improving load balancing and expert diversity.

**MoE in Open-Source Models.** MoE designs have been widely adopted in recent open-source large language models, though routing and balancing strategies vary considerably. DeepSeek-V3 (DeepSeek-AI, 2024b) proposes a bias-based, auxiliary-loss-free strategy that dynamically adjusts per-expert biases to achieve balance without explicit loss terms, while DeepSeek-V2 (DeepSeek-AI, 2024a) uses dual balancing objectives (expert utilization and token allocation) and device-limited routing to minimize communication costs. Qwen3-MoE (Yang et al., 2025) scales this paradigm with 128 experts and 8 active per token, introducing global-batch load balancing to aggregate statistics across devices for smoother gradients. OLMoE (Muennighoff et al., 2024) provides a fully reproducible baseline, combining Top-$k$ token-choice routing, Switch-style LBL, and a router z-loss for logit stabilization. LLaMA-MoE (Zhu et al., 2024) similarly applies dropless Top-$k$ routing with Switch-style balancing. In contrast, models like Mixtral-8×7B/22B (Jiang et al., 2024), GPT-OSS (Agarwal et al., 2025), DBRX (Databricks, 2024), and Grok-2 (gro, 2025) disclose expert counts and Top-$k$ settings but do not specify loss formulations, suggesting reliance on Switch-derived techniques.

## 3 APPROACH

In this section, we present GatePro, a parameter-free approach for improving expert selection diversity in Mixture-of-Experts models. We first provide the mathematical formulation of conventional MoE layers, then introduce our gate similarity computation and localized ompetition mechanism, as shown in Figure 1.

### 3.1 PRELIMINARIES: MIXTURE-OF-EXPERTS LAYER

Consider a standard MoE layer with $N$ experts $\{E_1, E_2, \ldots, E_N\}$, where each expert $E_i$ is typically a feed-forward network. We will use $[N]$ to denote the index set $\{1, 2, \ldots, N\}$. Given an input token $\mathbf{x} \in \mathbb{R}^d$, the router produces expert logits:

$$\boldsymbol{\ell}(\mathbf{x}) := \mathbf{W}_g \cdot \mathbf{x} + \mathbf{b}_g \in \mathbb{R}^N, \tag{1}$$

where $\mathbf{W}_g \in \mathbb{R}^{N \times d}$ and $\mathbf{b}_g \in \mathbb{R}^N$. Through this paper, we set $\mathbf{b}_g \equiv 0$ and use $\boldsymbol{\ell}_i(\cdot)$ to denote the $i$-th component of this function. We first select the top-$k$ expert subset *by logits*:

$$\mathcal{T}_k\big(\boldsymbol{\ell}(\mathbf{x})\big) := \underset{I \subset [N], |I| = k}{\arg\max} \sum_{i \in I} \boldsymbol{\ell}_i(\mathbf{x}). \tag{2}$$

Then we normalize *only over the selected set* to get mixture weights:

$$\alpha_i(\mathbf{x}) := \begin{cases} \dfrac{\exp\big(\boldsymbol{\ell}_i(\mathbf{x})\big)}{\sum_{j \in \mathcal{T}_k(\boldsymbol{\ell}(\mathbf{x}))} \exp\big(\boldsymbol{\ell}_j(\mathbf{x})\big)}, & i \in \mathcal{T}_k\big(\boldsymbol{\ell}(\mathbf{x})\big), \\ 0, & i \notin \mathcal{T}_k\big(\boldsymbol{\ell}(\mathbf{x})\big). \end{cases} \tag{3}$$

The MoE output is the sparsely weighted combination of expert outputs:

$$\mathbf{y} := \sum_{i \in [N]} \alpha_i(\mathbf{x}) \cdot E_i(\mathbf{x}). \tag{4}$$

### 3.2 GATEPRO APPROACH

GatePro addresses the expert selection problem by pronouncing both functional diversity and load balance. While existing methods treat expert selection as a token distribution challenge, GatePro recognizes that the critical issue is functional redundancy among co-selected experts. By introducing localized competition between the most similar expert pairs, GatePro encourages diversity in expert

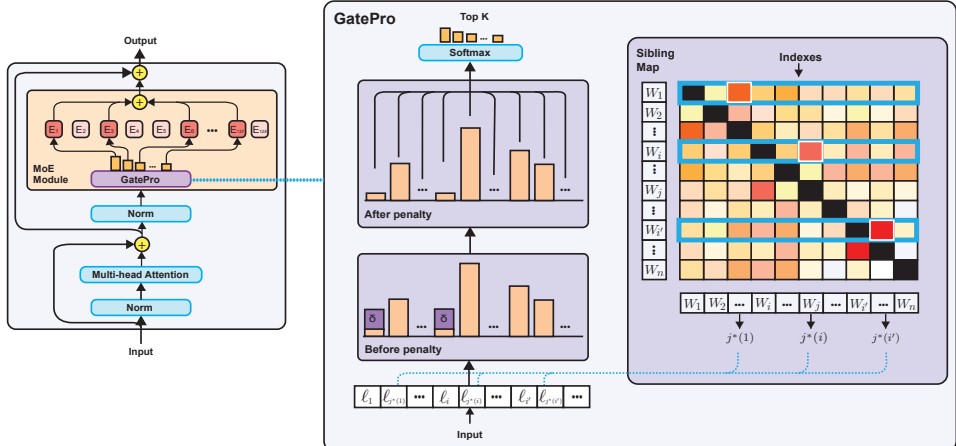

Figure 1: The GatePro Approach.

selection while maintaining the original model capacity. Unlike existing MoE optimization methods that rely on auxiliary losses or additional parameters, GatePro operates through a competitive propagation mechanism that directly prevents functionally similar experts from being co-activated.

**Gate Similarity Computation.** The first step in GatePro is to identify which expert pairs are most likely to provide redundant functionality. We conceptualize this as analyzing how expert specializations propagate through their gating patterns. Experts with similar gating weight vectors tend to be activated by similar types of tokens, indicating that their learned specializations have propagated along similar directions in the parameter space, potentially leading to functional overlap.

We define the cosine similarity matrix of the gating weights $\mathbf{S} \in \mathbb{R}^{n \times n}$ as:

$$S_{ij} := \frac{\langle \mathbf{w}_{g,i}, \mathbf{w}_{g,j} \rangle}{|\mathbf{w}_{g,i}| \cdot |\mathbf{w}_{g,j}|} \tag{5}$$

where $\mathbf{w}_{g,i}$ and $\mathbf{w}_{g,j}$ are the $i$-th and $j$-th rows of the gating weight matrix $\mathbf{W}_g$. This similarity matrix captures how specialization patterns have propagated across experts during training, with values ranging from -1 to 1. Higher similarity values indicate that experts have developed along similar specialization trajectories, suggesting potential redundancy that requires competitive selection.

**Localized Competition Mechanism.** Once we have identified similar expert pairs, the next challenge is promoting expert diversity without disrupting natural specialization patterns. We introduce localized competition between the most similar expert pairs, rather than applying global constraints that may interfere with all experts. The key insight of localized competition is that if two experts exhibit highly similar gating patterns, allowing both to be selected simultaneously provides diminishing returns and creates redundancy. Therefore, we encourage competitive selection that favors the expert with stronger activation for the current token, promoting enhanced expert diversity.

For each expert $i$, we identify its most similar counterpart:

$$j^*(i) := \arg\max_{j \neq i} S_{ij} \tag{6}$$

The competition winner is determined based on the gating probabilities for the current input token:

$$\text{winner}(i, j^*(i))[\mathbf{x}] := \begin{cases} i & \text{if } \boldsymbol{\ell}_i(\mathbf{x}) \geq \boldsymbol{\ell}_{j^*(i)}(\mathbf{x}) \\ j^*(i) & \text{otherwise} \end{cases} \tag{7}$$

This token-specific competition ensures that the selection depends on the actual relevance of each expert for the current input. The losing expert is suppressed by applying a constant negative penalty to its logit:

$$\tilde{\boldsymbol{\ell}}_i(\mathbf{x}) := \begin{cases} \boldsymbol{\ell}_i(\mathbf{x}) & \text{if } \text{winner}(i, j^*(i))[\mathbf{x}] = i \\ \boldsymbol{\ell}_i(\mathbf{x}) - \lambda & \text{if } \text{winner}(i, j^*(i))[\mathbf{x}] = j^*(i) \end{cases} \tag{8}$$

where $\lambda$ is a positive constant (typically $\lambda = 10^{-4}$). This aggressive penalty mechanism effectively eliminates the losing expert from consideration while maintaining numerical stability. Then we

compute the mixture weights by the suppressed logits $\tilde{\ell}(\mathbf{x})$:

$$\tilde{\alpha}_i(\mathbf{x}) := \begin{cases} \dfrac{\exp(\tilde{\ell}_i(\mathbf{x}))}{\sum_{j \in \mathcal{T}_k((\tilde{\ell}(\mathbf{x}))} \exp(\tilde{\ell}_j(\mathbf{x}))}, & i \in \mathcal{T}_k(\tilde{\ell}(\mathbf{x})), \\ 0, & i \notin \mathcal{T}_k(\tilde{\ell}(\mathbf{x})). \end{cases} \tag{9}$$

The final output is computed as follows:

$$\tilde{\mathbf{y}} := \sum_{i \in [N]} \tilde{\alpha}_i(\mathbf{x}) \cdot E_i(\mathbf{x}). \tag{10}$$

where $\tilde{\mathbf{y}}$ is the final output that benefits from improved expert diversity through the GatePro competition mechanism. The GatePro approach is summarized in Algorithm 1. The computational overhead is minimal compared to the base MoE computation. The cosine similarity matrix computation has $O(N^2 d)$ complexity. The competitive selection requires only $O(N)$ complexity per token. Without requiring additional parameters, GatePro can be easily integrated into existing MoE architectures. Unlike auxiliary loss methods that require careful tuning and may interfere with training, GatePro can be enabled or disabled during training without additional learnable parameters, which we call hot-swappable. This feature allows flexible deployment and creates persistent improvements that benefit the model even after GatePro is disabled. More detailed analysis is shown in Section B.

---

**Algorithm 1** GatePro Approach

---

**Require:** Input token $\mathbf{x}$, gating weights $\mathbf{W}_g$, penalty $\lambda$, experts $\{E_1, \ldots, E_N\}$
**Ensure:** Final MoE output $\tilde{\mathbf{y}}$
1: Compute original logits: $\boldsymbol{\ell} = \mathbf{W}_g \mathbf{x}$
2: Compute gate similarity matrix: $\mathbf{S}$ by equation 5
3: Find most similar pairs: $j^*(i) = \arg\max_{j \neq i} \mathbf{S}_{ij}$ for each $i$
4: Initialize penalty mask: $\boldsymbol{\delta} = \mathbf{0}$
5: **for** each expert $i \in \{1, 2, \ldots, N\}$ **do**
6:    **if** $\boldsymbol{\ell}_i < \boldsymbol{\ell}_{j^*(i)}$ **then**
7:       $\delta_i = -\lambda$
8:    **end if**
9: **end for**
10: Apply penalties: $\tilde{\boldsymbol{\ell}} = \boldsymbol{\ell} + \boldsymbol{\delta}$
11: Select top-$k$ experts: $\mathcal{T}_k(\tilde{\boldsymbol{\ell}}(\mathbf{x}))$
12: Compute probabilities: $\tilde{\alpha}(\mathbf{x})$ by equation 9
13: Compute final output: $\tilde{\mathbf{y}} = \sum_{i \in [N]} \tilde{\alpha}_i(\mathbf{x}) \cdot E_i(\mathbf{x})$
14: **return** $\tilde{\mathbf{y}}$

---

## 4 EXPERIMENTS

In this section, we evaluate the effectiveness of GatePro across multiple model scales and benchmarks. We conduct comprehensive experiments on various downstream tasks to demonstrate the consistent improvements achieved by our parameter-free approach.

### 4.1 EXPERIMENTAL SETUP

We conduct GatePro experiments on two different model scales: MoE-0.7B/7B and MoE-1.3B/13B. These in-house MoE models are variants of the OLMoE architecture (Muennighoff et al., 2024) with increased training parameters, different activation parameters, and expert pool configurations. Both models use sparse MoE architectures with top-$k$ expert selection where $k = 6$. We train models from scratch using the same training configurations to ensure a fair comparison between baseline MoE models and GatePro. We systematically track model performance at multiple training intervals ranging from early (100B tokens) to advanced training stages (up to 1.2T tokens) to deeply understand the impact of GatePro throughout training. The training utilized distributed computing across 8 nodes with a total of 64 GPUs. The training incorporated advanced optimization techniques including FSDP (Zhao et al., 2023) and Flash Attention (Dao et al., 2022).

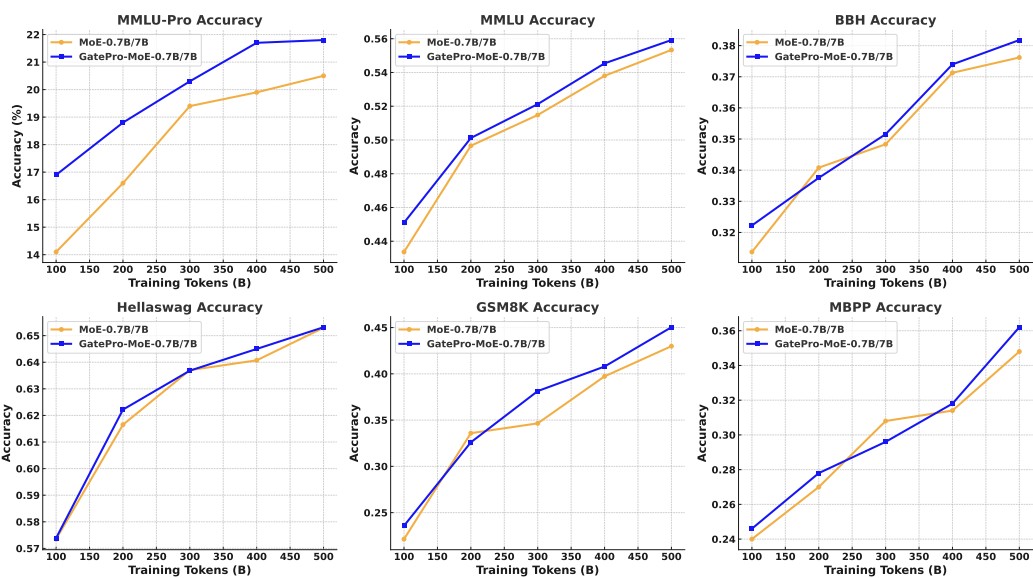

Figure 2: Performance comparsion on MoE-0.7B/7B pretrain stage.

The evaluation is conducted on six diverse benchmarks covering factual knowledge (MMLU-Pro (Wang et al., 2024) and MMLU (Hendrycks et al., 2020)), knowledge and commonsense reasoning (BBH (Suzgun et al., 2022) and HellaSwag (Zellers et al., 2019)), arithmetic reasoning (GSM8K (Cobbe et al., 2021)), and code generation (MBPP (Austin et al., 2021)).

## 4.2 MAIN RESULTS

Figure 2 and Figure 5 show that GatePro consistently outperforms baseline models across different model scales and training stages, including both pretraining and Continuous Training (CT). GatePro delivers substantial early-stage gains, faster convergence, and higher final accuracy without additional parameters beyond minimal gate similarity computations, validating its practical applicability.

GatePro demonstrates immediate and sustained advantages throughout the entire pretraining process. Notably, GatePro shows clear improvements as early as 100B tokens across all evaluated benchmarks and maintains these gains through training completion. At the MoE-0.7B/7B scale, MMLU-Pro achieves $16.9\%$ compared to baseline's $14.1\%$ at 100B tokens, with the advantage persisting to $21.8\%$ vs. $20.5\%$ at 500B tokens. Similarly, GSM8K demonstrates early improvements with GatePro achieving $23.6\%$ compared to baseline's $22.1\%$ at 100B tokens, expanding to $45.0\%$ vs. $43.0\%$ at 500B tokens. Additional benchmarks confirm this pattern: MMLU exhibits $1.7\%$ early gains that persist with $0.6\%$ advantage at 500B tokens, while BBH shows $0.8\%$ improvements at both stages. These results strongly suggest that GatePro effectively achieves enhanced expert selection diversity, enabling more efficient use of model capacity throughout the entire training process.

When scaling to MoE-1.3B/13B, GatePro's advantages persist and further expand. Figure 5 shows that GSM8K improves from $64.7\%$ to $65.5\%$ at 1.2T tokens, while BBH rises from $49.8\%$ to $50.7\%$, highlighting GatePro's capacity to mitigate gate redundancy even at large scales. These results demonstrate that reasoning-intensive tasks particularly benefit from diversified expert selection, while factual knowledge tasks like MMLU and HellaSwag show more incremental gains across all training stages. Comprehensive results across expert pool sizes are provided in Section A.2.

We further examine GatePro's effectiveness at the Continuous Training (CT) stage as summarized in Table 1. At MoE-0.7B/7B, GatePro improves overall performance from $51.92\%$ to $52.55\%$, with particularly strong gains in arithmetic reasoning and knowledge tasks. Scaling to MoE-1.3B/13B, GatePro enhances performance from $63.95\%$ to $64.88\%$, demonstrating consistent improvements across various tasks. More detailed comparison is provided in Section A.4.

| Model | Benchmarks | | | | | | Overall |
|---|---|---|---|---|---|---|---|
| | MMLU-Pro | MMLU | BBH | HellaSwag | GSM8K | MBPP | |
| 0.7B/7B MoE | 30.7 | 62.2 | 46.7 | 66.4 | 63.3 | 42.2 | 51.92 |
| **GatePro-MoE** | **31.4** | **62.1** | **47.2** | **66.4** | **65.2** | **43.0** | **52.55** |
| 1.3B/13B MoE | 41.8 | 71.7 | 62.5 | 73.6 | 77.7 | 56.4 | 63.95 |
| **GatePro-MoE** | **42.2** | **72.0** | **62.5** | **74.6** | **79.7** | **58.3** | **64.88** |

Table 1: Performance Comparison on CT stage. Best results are highlighted in **blue**.

## 4.3 COMPARISION WITH OLMoE

To further validate GatePro's generalizability, we extend evaluation to the open-source OLMoE-1B/7B architecture (Muennighoff et al., 2024), as presented in Table 2. This experimental setup follows the original open-source OLMoE configuration exactly without any architectural modifications. This analysis serves to demonstrate that GatePro's effectiveness applies across both in-house MoE architectures and widely-adopted MoE implementations in the research community.

The results demonstrate consistent improvements across all evaluated benchmarks. For knowledge tasks, MMLU shows an improvement from 37.6% to 38.3%. The ARC-Challenge benchmark exhibits notable enhancement, achieving a 1.1% improvement that validates effective expert specialization for knowledge-intensive queries. Commonsense reasoning capabilities are enhanced across multiple tasks, with PIQA improving by 0.82% and COPA advancing by 0.7%. Even the challenging HellaSwag benchmark shows stable enhancement with a 0.4% improvement that validates GatePro's consistent benefits across varying degrees of expert redundancy. Furthermore, the overall performance metric demonstrates a substantial 0.7% absolute improvement across the benchmarks. The improvements demonstrate GatePro's robustness when applied to different architectural foundations and confirm its ability to improve expert diversity in diverse reasoning scenarios. The OLMoE experimental results demonstrate that GatePro approach is effective across diverse MoE implementations, establishing it as a broadly applicable solution for improving expert utilization.

| Model | Benchmarks | | | | | Overall |
|---|---|---|---|---|---|---|
| | MMLU | HellaSwag | ARC-Challenge | PIQA | COPA | |
| OLMoE-1B-7B | 37.6 | 69.0 | 39.46 | 76.87 | 86.0 | 61.8 |
| OLMoE-GatePro-1B-7B | **38.3** | **69.4** | **40.57** | **77.69** | **86.7** | **62.5** |

Table 2: Performance comparison on OLMoE (400B tokens). Best results are highlighted in **blue**.

## 5 EXPERT UTILIZATION ANALYSIS

To validate our hypothesis that GatePro promotes more efficient expert utilization and enhanced expert selection diversity, we track expert activation patterns throughout training by monitoring *zero token counts*: the number of experts receiving zero tokens at each training step. This metric serves as a direct indicator of expert underutilization. Figure 3 shows the progression of *zero token counts* across six representative layers for four configurations: Baseline without balance loss, GatePro without balance loss, Baseline, and GatePro. We analyze the results from two key perspectives: Accelerated Expert Activation and Layer-Dependent Utilization Patterns.

**Accelerated Expert Activation.** GatePro accelerates expert activation across all network layers. In shallow layers, while all configurations eventually converge to near-zero unused experts, GatePro consistently exhibits steeper initial decline curves. For instance, in Layer 7, GatePro without balance loss reduces the number of unused experts from 128 to 20 within the first 1500 steps, whereas the baseline without balance loss requires nearly 2500 steps to achieve a similar reduction. This demonstrates that GatePro achieves superior load balancing performance through its diversity-driven competitive propagation mechanism.

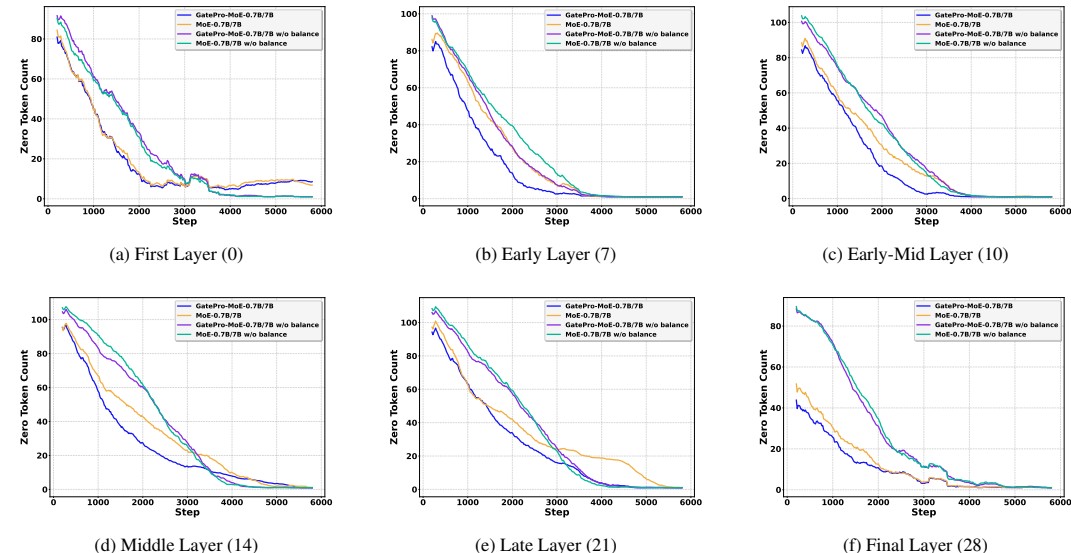

Figure 3: Zero token count progression across different layers during training with 128 experts. The figure shows comparison between different configurations across shallow and deep layers: MoE w/o balance loss (green), GatePro w/o balance loss (purple), MoE (orange), and GatePro (blue).

After applying balance loss, both baseline and GatePro show improved convergence, with GatePro achieving faster convergence - reaching 20 unused experts in only 1000 steps, compared to 2000 steps for baseline. The acceleration advantage is amplified in middle and deep layers. In Layer 14, GatePro reduces unused experts from 128 to 20 in approximately 1500 steps, while baselines require 3000 steps. These results indicate that GatePro and balance loss complement each other in MoE models rather than operating redundantly. Similar acceleration patterns appear consistently in deeper layers, demonstrating that GatePro's competitive propagation mechanism maintains effectiveness across different network depths throughout the architecture. This accelerated activation demonstrates enhanced specialization, where experts develop more distinct capabilities.

**Layer-Dependent Utilization Patterns.** We observe that deeper layers require significantly more training steps to achieve full expert utilization compared to shallow layers. Shallow layers typically reach near-zero unused experts within 4000 steps, while deeper layers require more steps to achieve similar utilization levels. This depth-dependent activation delay suggests that expert specialization in deeper layers is inherently more challenging, as these layers must learn more complex and abstract representations that require longer training periods to establish clear functional boundaries between experts. This pattern holds consistently across different expert pool sizes. As shown in Figure 6, when we scale from 128 to 256 experts, we observe similar depth-dependent behaviors: shallow and middle layers show relatively modest differences between configurations. However, in deeper layers, the convergence becomes significantly slower, requiring 10000 steps for complete activation. GatePro maintains its acceleration advantages across both 128 and 256 expert configurations, demonstrating effective scaling with increased expert pool sizes. This reinforces GatePro's importance in deeper layers where baseline methods face the greatest activation challenges.

## 6 EXPERT GATING SIMILARITY ANALYSIS

To investigate GatePro's mechanism for improving expert selection diversity, we analyze expert gating similarity patterns across four metrics for MoE with 128 experts in shallow and deep layers, as shown in Figure 4. The definitions of metrics and the analysis for MoE with 256 experts are postponed in Section C. This analysis examines how GatePro achieves improved complementarity where experts become complementary rather than redundant. Lower cosine similarity, higher angles, and higher entropy indicate expert diversity and reduced redundancy.

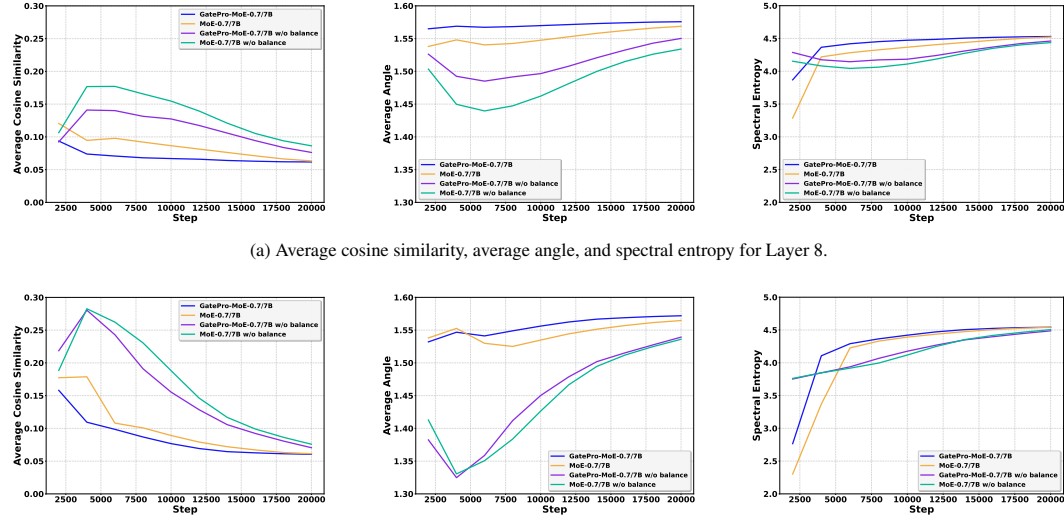

(a) Average cosine similarity, average angle, and spectral entropy for Layer 8.

(b) Average cosine similarity, average angle, and spectral entropy for Layer 16.

Figure 4: Expert gating similarity analysis for MoE with 128 experts. Metrics at Layer 8 and Layer 16. Each row shows four metrics: average cosine similarity, average angle, and spectral entropy.

**Average Cosine Similarity.** GatePro consistently maintains lower cosine similarity values across both layers, demonstrating superior expert diversity compared to baseline configurations. Configurations without balance loss show higher similarity values, indicating worse diversity. The sustained low values throughout GatePro training validate that the competitive propagation mechanism successfully encourages distinct expert specializations. This reduction in similarity is particularly important for preventing functional redundancy in expert selection.

**Average Angle.** Both layers exhibit larger average angles for GatePro, indicating better expert differentiation throughout training. Configurations without balance loss demonstrate lower angular separation, suggesting reduced differentiation. The consistently higher angular separation demonstrates that GatePro maintains meaningful distinctions between experts, improving expert diversity and ensuring unique capabilities rather than redundant functionality.

**Spectral Entropy.** Expert selection entropy demonstrates substantial improvements across both layers, with GatePro achieving higher entropy values that confirm more balanced and uniform expert utilization. Without balance loss configurations show lower entropy values, indicating less balanced expert utilization. Higher entropy indicates that the gating mechanism distributes computational load more evenly across all available experts, preventing the concentration of activations on a subset of dominant experts and maximizing the model's representational capacity. These consistent improvements across all metrics provide compelling evidence that GatePro effectively improve expert selection diversity throughout the entire architecture. The results demonstrate both enhanced specialization through lower cosine similarity and higher angular separation, and improved complementarity via higher entropy values.

## 7 CONCLUSION

We propose GatePro, a novel parameter-free approach that addresses expert selection diversity in Mixture-of-Experts models through competitive propagation mechanisms. Unlike conventional balance loss methods that focus on statistical load distribution, GatePro prevents functionally similar experts from being simultaneously selected through localized competition, directly addressing expert underutilization. Our experimental evaluation demonstrates GatePro's consistent effectiveness across multiple model scales and benchmarks, with particularly strong improvements in deeper layers where expert specialization is most challenging. The analysis reveals enhanced specialization and improved complementarity, with accelerated expert activation and superior diversity metrics. GatePro's parameter-free design enables flexible deployment for real-world applications while establishing an effective foundation for MoE optimization that considers both diversity and balance.

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

## A APPENDIX

### A.1 USAGE OF LLM.

We utilized ChatGPT to improve the manuscript's readability.

### A.2 LARGE-SCALE MODEL EVALUATION

At the larger MoE-1.3B/13B scale, GatePro maintains consistent improvements across all benchmarks throughout the extended training period. MMLU-Pro demonstrates steady gains, with baseline achieving 24.2% compared to GatePro's 25.6% at 300B tokens, and the advantage persisting with baseline at 30.6% vs. GatePro's 31.6% at 1.2T tokens. Complex reasoning tasks show particularly strong benefits, with BBH improving from baseline's 41.8% to GatePro's 42.3% at 300B tokens and expanding to 49.8% vs. 50.7% at 1.2T tokens. GSM8K exhibits similar patterns, with baseline achieving 52.5% compared to GatePro's 52.6% at 300B tokens, growing to 64.7% vs. 65.5% at 1.2T tokens. These results confirm that GatePro's benefits scale effectively with model size and training duration, demonstrating robust performance improvements across diverse task categories.

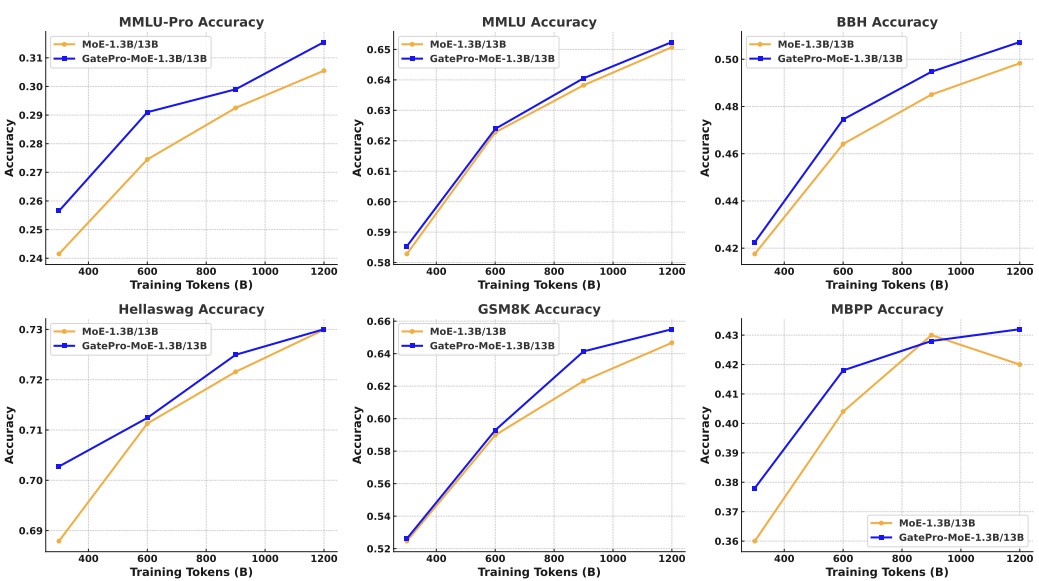

Figure 5: Performance Comparsion on in-house MoE-1.3B/13B pretrain stage.

### A.3 ZERO TOKEN COUNT PROGRESSION ACROSS 256 EXPERTS

When scaling to 256 experts, GatePro's advantages become even more pronounced across all network layers. The increased expert pool size creates greater challenges for efficient utilization, yet GatePro consistently demonstrates superior convergence rates compared to baseline configurations. In shallow layers such as Layer 0 and Layer 7, GatePro configurations (both with and without balance loss) achieve faster reduction in unused experts, with steeper decline curves that reach near-zero unused experts more rapidly than their baseline counterparts.

The benefits are particularly striking in deeper layers, where the complexity of expert specialization typically leads to slower activation patterns. In Layer 21 and Layer 28, GatePro maintains its acceleration advantage even with the expanded 256-expert pool, demonstrating that the competitive propagation mechanism scales effectively with increased expert capacity. Notably, the combination of GatePro with balance loss achieves the most rapid convergence across all layers, suggesting optimal synergy between diversity-driven competition and load balancing mechanisms.

These results with 256 experts validate that GatePro's effectiveness is not limited by expert pool size, but rather becomes more valuable as the number of available experts increases, addressing the growing challenge of efficient expert utilization in large-scale MoE architectures.

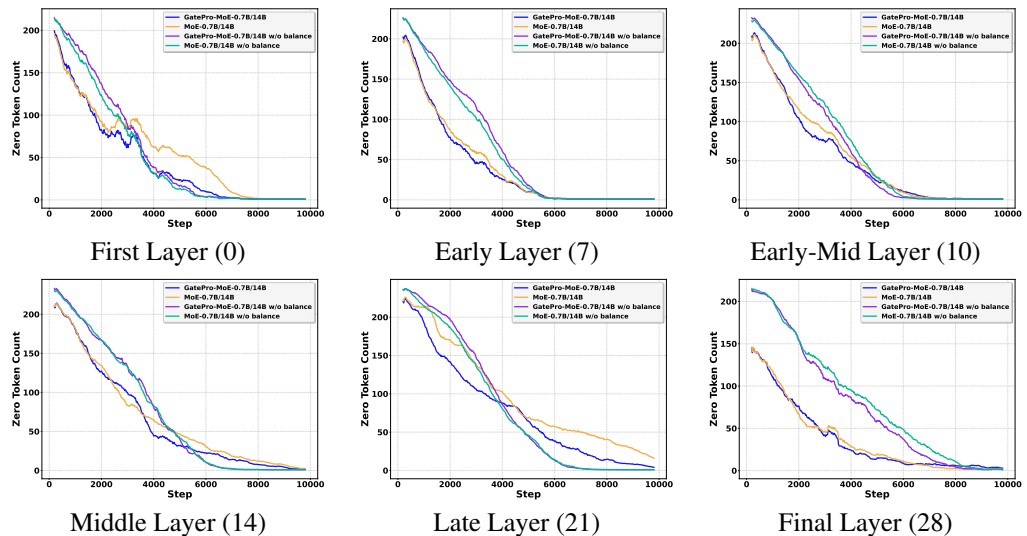

Figure 6: Zero token count progression across different layers during training with 256 experts. The figure shows the comparison between different training configurations across six representative layers spanning the entire network depth: Baseline w/o balance (green), GatePro w/o balance (purple), GatePro with balance (blue), and Baseline with balance (orange).

### A.4 GATEPRO PERFORMANCE ANALYSIS ON MOE CT STAGE

We further examine GatePro's efficacy through detailed performance comparisons at the Continuous Training (CT) stage for MoE models at MoE-0.7B/7B and MoE-1.3B/13B, as summarized in Table 1. At the MoE-0.7B/7B, GatePro yields overall improvement from $51.92\%$ to $52.55\%$, with particularly strong gains in arithmetic reasoning (GSM8K) and knowledge reasoning tasks (BBH). The method also provides meaningful improvements across factual knowledge and code generation tasks, highlighting its broad applicability.

Scaling to MoE-1.3B/13B, GatePro continues to demonstrate notable improvements, enhancing overall performance from $63.95\%$ to $64.88\%$. The improvements are particularly substantial in arithmetic reasoning and coding generation tasks, confirming GatePro's ability to effectively leverage increased model capacity. GatePro also achieves consistent gains across semantic memory and commonsense inference tasks, underscoring its robustness in enhancing expert diversity across multiple cognitive domains.

## B HOT-SWAPPABLE TRAINING ANALYSIS

To validate GatePro's "hot-swappable" deployment flexibility, we conducted experiments with different training phase configurations, where models transition between GatePro-MoE and MoE during training. Table 3 presents performance results across different switching schedules using the GatePro-MoE 0.7B/14B architecture with 256 experts.

The results reveal a clear trend: longer initial training with GatePro leads to progressively better final performance. The configuration with 400B tokens of GatePro training followed by 100B tokens of standard training achieves the best performance on BBH (44.5%) and MBPP (45.5%), while the full 500B GatePro training achieves the highest scores on MMLU-Pro (30.1%), MMLU (63.4%), and GSM8K (42.0%). This pattern suggests that GatePro's diversity benefits accumulate over training, with longer exposure to competitive propagation leading to better expert specialization.

| Training Configuration | MMLU-Pro | MMLU | BBH | GSM8K | MBPP |
|---|---|---|---|---|---|
| 100B GatePro-MoE → 400B MoE | 28.7 | 61.4 | 43.0 | 40.9 | 43.1 |
| 200B GatePro-MoE → 300B MoE | 29.0 | 62.5 | 43.4 | 41.3 | 43.1 |
| 300B GatePro-MoE → 200B MoE | 29.7 | 63.1 | 43.8 | 41.6 | 44.7 |
| 400B GatePro-MoE → 100B MoE | 30.0 | 63.2 | **44.5** | 41.6 | **45.5** |
| 500B GatePro-MoE (Full) | **30.1** | **63.4** | 44.2 | **42.0** | 44.9 |

Table 3: Performance comparison across GatePro-MoE 0.7B/14B training schedules with 256 experts. Arrow notation indicates training phase transitions (e.g., "100B GatePro-MoE → 400B MoE" means training with GatePro for the first 100B tokens, then disabling GatePro and continuing training with standard MoE for the remaining 400B tokens).

These findings validate GatePro's practical value for real-world deployment scenarios. Organizations can strategically apply GatePro during computationally intensive early training phases to establish good expert diversity, then switch to standard training for resource efficiency without sacrificing performance gains. The parameter-free nature ensures that such transitions require no architectural modifications or hyperparameter retuning, making deployment decisions purely operational rather than technical.

The results demonstrate that GatePro's competitive propagation mechanism creates persistent improvements in expert utilization patterns that continue to benefit the model even after the mechanism is disabled. This "training legacy effect" makes GatePro particularly valuable for practitioners seeking to optimize training efficiency while maintaining model quality across different deployment constraints.

## C  EXTENDED GATING SIMILARITY ANALYSIS

In this section, we provide precise definitions of the evaluation metrics used in our analysis and present additional results from runs with 256 experts. These metrics are designed to capture different aspects of expert diversity and specialization within the mixture-of-experts layer. Formally, we define the following:

- **Average Cosine Similarity.** This metric measures the overall alignment between expert gating vectors. It is computed as the mean absolute cosine similarity across all pairs of experts:

$$\text{Average cosine similarity} := \frac{2}{N(N-1)} \sum_{1 \leq i < j \leq N} |S_{ij}|.$$

Lower values indicate that experts tend to activate on different tokens, while higher values suggest stronger redundancy.

- **Average Angle.** Complementary to cosine similarity, we also compute the average angle between experts:

$$\text{Average angle} := \frac{2}{N(N-1)} \sum_{1 \leq i < j \leq N} \arccos(S_{ij}).$$

A larger average angle indicates greater orthogonality between expert behaviors, whereas smaller angles correspond to more overlapping activation patterns.

- **Spectral Entropy.** To capture the diversity of expert activations at a more global scale, we consider the entropy of the singular values of the similarity matrix $\mathbf{S}$. Let $\sigma_1, \sigma_2, \ldots, \sigma_N$ denote the singular values. We normalize them by:

$$\tilde{\sigma}_i := \frac{\sigma_i + \epsilon}{\sum_{i \in [N]} \sigma_i + N \cdot \epsilon}, \qquad \epsilon = 10^{-8},$$

and define the entropy as

$$\text{Spectral entropy} := - \sum_{i \in [N]} \tilde{\sigma}_i \log \tilde{\sigma}_i.$$

Intuitively, this metric reflects how evenly spread the similarity spectrum is: higher entropy implies more balanced expert specialization, while lower entropy suggests that only a few dominant modes exist.

For average cosine similarity and spectral entropy, larger values indicate that expert directions are more dispersed, which corresponds to better diversity. In contrast, for average angle, smaller values imply the same effect. Consistent with the patterns we observed earlier in Fig. 4, the 256-expert results in Fig. 7 highlight two key trends:

- **Balanced expert utilization.** GatePro achieves more uniform and equitable distribution of tokens across experts compared to the baseline, preventing collapse where only a few experts dominate.

- **Sharp and concentrated similarity distribution.** GatePro produces histograms with sharper peaks concentrated near zero similarity, whereas models trained without the balance loss exhibit skewed and unstable distributions, reflecting poor expert diversification.

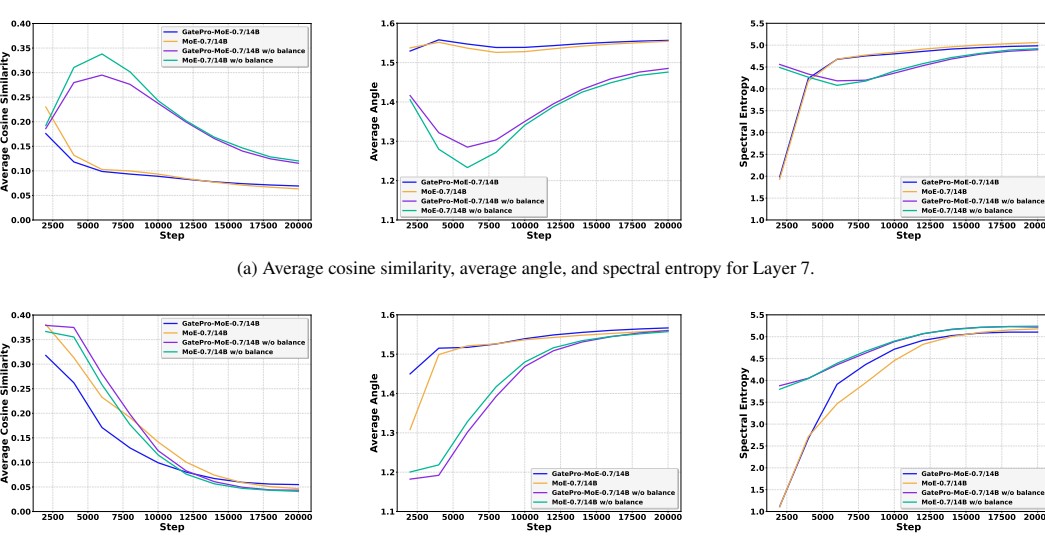

(a) Average cosine similarity, average angle, and spectral entropy for Layer 7.

(b) Average cosine similarity, average angle, and spectral entropy for Layer 17.

Figure 7: Expert gating similarity analysis for MoE with 256 experts. Metrics at Layer 7 and Layer 17. Each row shows four metrics: average cosine similarity, average angle, and spectral entropy.

