# OpenReview forum: "GatePro: Parameter-Free Expert Selection Optimization for Mixture-of-Experts Models"
_ICLR.cc/2026/Conference — Submitted to ICLR 2026_

### Official Review · Reviewer_F5vF · 2025-10-28

**Soundness:** 3
**Presentation:** 3
**Contribution:** 3
**Rating:** 4
**Confidence:** 4

**Summary:**

This paper introduces GatePro, a parameter-free method that addresses expert selection diversity in Mixture-of-Experts models by introducing localized competition between functionally similar expert pairs. The approach identifies similar experts via cosine similarity and applies penalties to prevent their co-activation.

**Strengths:**

This paper identifies the expert selection diversity problem in MoE architectures, and proposes a simple yet effective solution that requires no additional learnable parameters. The experimental evaluation is comprehensive, covering multiple model scales, training stages, and benchmarks. The hot-swappable design offers practical value for real-world deployment.

**Weaknesses:**

(1) While the paper presents an intuitive motivation for introducing competition between similar experts, it lacks analysis to explain why cosine similarity of gating weights is the optimal metric for identifying functionally redundant experts. Two experts might have similar gating weight directions but could still process tokens differently due to their internal FFN parameters. For example, two experts might both be activated by technical tokens (leading to similar gating weights), but one could specialize in mathematical reasoning while the other handles programming syntax.

(2) The paper introduces a fixed penalty coefficient λ = 10⁻⁴ for the losing expert in the competition mechanism, but provides limited investigation of its sensitivity. Similarly, the paper uses k=6 active experts across all experiments, but doesn't explore how GatePro's effectiveness might vary with different k values.

(3) The paper demonstrates that GatePro reduces expert similarity and increases diversity metrics, but it provides limited insight into what kinds of specializations the experts actually learn. Do experts under GatePro develop more interpretable specializations?

(4) The paper identifies the most similar expert j*(i) for each expert i based on gating weight similarity, but does not clearly specify how often this mapping is updated during training. During training, experts continuously learn and specialize, causing their similarity relationships to change. If the mapping is infrequently updated, it may not adapt to the evolving specialization of experts during training. If recomputed at every step, this adds computational overhead that is not accounted for.

**Questions:**

The experimental evaluation primarily compares GatePro against baseline MoE with standard balance loss, but does not compare against other potential diversity-promoting approaches mentioned in related work. How does GatePro compare to other methods such as entropy regularization, orthogonality constraints, or decorrelation losses?

---

> ### Author Response · Authors · 2025-11-21
>
> Thank you very much for your detailed and constructive reviews. Your comments greatly help us improve the paper. Our responses are provided below.
>
> **Answer to Question 1**: Gating weights determine the token-to-expert routing policy, which defines each expert's operational domain in the input space. When two experts have high cosine similarity in their gating weights, they respond similarly to the same input distributions, meaning they compete for the same tokens regardless of their internal FFN parameters. This creates redundancy at the routing level - even if their FFN transformations differ, consistently co-activating these experts on similar inputs wastes model capacity that could be used to cover different parts of the input space.
>
> Our empirical results validate this approach: GatePro achieves consistent improvements across diverse tasks, model scales (7B to 13B), and training stages. The average cosine similarity analysis in figure 4 and the improvements in entropy, angular separation, and performance metrics demonstrate that gating weight similarity effectively identifies redundant patterns. Regarding the technical token example: if experts truly specialize differently (math vs. programming), their gating weights should naturally diverge during training. High gating similarity indicates insufficient specialization, which is precisely the redundancy GatePro addresses.
>
> **Answer to Question 2**: Based on your question, we conducted additional ablation experiments using  λ = 1e-1, 1e-2, 1e-3, and 1e-4 on 256 experts and compared performance on MMLU-Pro, MMLU, BBH, HellaSwag, GSM8K, and MBPP with 256 experts:
>
> |  λ  | MMLU-Pro | MMLU | BBH  | HellaSwag | GSM8K | MBPP |
> | ---- | ---- | ---- | ---- | ---- | ---- | ---- |
> | GatePro-MoE 1e-1     | 31.9     | 65.3 | 50.9 | 73.1      | 65.6  | 43.5 |
> | GatePro-MoE 1e-2     | 31.6     | 65.4 | 50.6 | 73.0      | 65.6  | 43.3 |
> | GatePro-MoE 1e-3     | 31.5     | 65.3 | 50.7 | 73.1      | 65.4  | 43.1 |
> | GatePro-MoE 1e-4     | 31.5     | 65.2 | 50.7 | 73.0      | 65.5  | 43.2 |
> | MoE baseline | 30.5     | 64.8 | 49.8 | 72.8      | 64.6  | 42.0 |
>
> As shown above, using a larger λ (1e−1) does produce slightly higher improvements compared to 1e−4. However, the improvements are relatively modest. In our paper, for 256 experts specifically, a larger λ indeed yields slightly better results, as your intuition suggests. We agree this is a meaningful ablation, and we will include this comparison and discussion in the revised version of the paper.
>
> The reason for choosing top-k = 6 is that, as shown in our paper, the two models used in our experiments have 0.7B/7B and 1.3B/13B active-to-total parameter ratios, which follow the principle that the number of active parameters is approximately one-tenth of the total parameters. If we change the value of top-k, the active-parameter budget would no longer match the 1/10 design, making the comparison inconsistent.
>
>
> **Answer to Question 3**: Understanding what specific functions individual experts learn is a challenging question about MoE interpretability that extends far beyond the scope of our work. This question requires dedicated research with specialized interpretability techniques such as expert activation analysis across diverse input types, neuron-level probing, and systematic ablation studies - essentially a separate research effort focused specifically on MoE interpretability.
>
> Our work focuses on a different but complementary objective: improving expert selection diversity at the routing level to enhance model performance. The consistent improvements across diverse benchmarks (reasoning, knowledge, coding) and the increases in diversity metrics (entropy, angular separation) demonstrate that GatePro successfully encourages experts to develop more distinct functions, even without explicitly analyzing what those functions are. We view expert specialization interpretability as an important future direction that would benefit the broader MoE community, but it requires methodological approaches and experimental designs beyond the scope of our performance-focused study.
>
> **Answer to Question 4**: We recompute the expert similarity mapping at every training step to ensure the competition mechanism adapts immediately to evolving expert specializations. While this might seem computationally expensive, our empirical measurements demonstrate that the overhead is negligible in practice.
>
> | Model        | tokens/day (B) |  MFU   |
> | ----- | -----  | ----- |
> | MoE baseline | 71.31          | 0.413 |
> | GatePro-MoE  | 71.24         |  0.411 |
>
> GatePro-MoE shows nearly identical throughput and comparable MFU to the baseline.
>
> Your review is extremely important to the quality of our work. We sincerely hope that the responses provided here address all your concerns, and we would be grateful if you could raise the score. Thank you very much.

---

### Official Review · Reviewer_bjg4 · 2025-10-30

**Soundness:** 3
**Presentation:** 3
**Contribution:** 3
**Rating:** 6
**Confidence:** 3

**Summary:**

This paper proposes GatePro, a parameter-free method to improve expert selection in MoE models by introducing localized competition between the most similar experts. The approach reduces redundant expert co-activation, improves expert diversity, and leads to consistent performance gains across model scales and benchmarks.

**Strengths:**

The method is parameter-free and can be added or removed during training without affecting the architecture, which makes it easy to deploy in practice.

The approach directly targets expert redundancy and shows consistent improvements across multiple scales and tasks, including both reasoning and knowledge benchmarks.

The analysis is thorough, including expert utilization, similarity metrics, and training-stage comparisons, which provides clear evidence that the method improves expert diversity.

**Weaknesses:**

The experiment lacks computational cost and stability analysis. Calculating the similarity matrix becomes burdensome as the number of experts increases, especially when expert parallelism is enabled, requiring frequent communication. Therefore, a quantitative analysis of training speed, memory usage, and communication costs is needed.

The experiment fixed the hyperparameter coefficients for the competition penalty; changing these coefficients may have different effects on model performance. More sensitivity analysis or theoretical justification is required.

**Questions:**

Refer to Weaknesses

---

> ### Author Response · Authors · 2025-11-21
>
> Thank you very much for your detailed and constructive reviews. Your review greatly helps us improve the paper. Our responses are provided below.
>
> **Answer to Question 1**: All experiments were trained under FSDP, and we measured throughput, memory usage, and compute efficiency under identical settings. The results are:
>
> | Model        | tokens/day (B) | max memory (GB) | MFU   |
> | ----- | -----  | -----  | ----- |
> | MoE baseline | 71.31          | 40.82           | 0.413 |
> | GatePro-MoE  | 71.24          | 40.93           | 0.411 |
>
> GatePro-MoE shows nearly identical throughput and comparable memory usage and MFU to the baseline. Since the similarity computation is lightweight and computed locally on each FSDP shard, it does not introduce any additional communication or synchronization beyond standard FSDP all-reduces.
>
> Therefore, GatePro shows no noticeable computational or communication overhead, and training remains fully stable in practice. We will provide the throughput, memory usage, and compute efficiency results in the appendix
>
> **Answer to Question 2**: Based on your question, we conducted additional ablation experiments using  λ = 1e-1, 1e-2, 1e-3, and 1e-4 on 256 experts and compared performance on MMLU-Pro, MMLU, BBH, HellaSwag, GSM8K, and MBPP with 256 experts:
>
> |  λ  | MMLU-Pro | MMLU | BBH  | HellaSwag | GSM8K | MBPP |
> | ---- | ---- | ---- | ---- | ---- | ---- | ---- |
> | GatePro-MoE 1e-1     | 31.9     | 65.3 | 50.9 | 73.1      | 65.6  | 43.5 |
> | GatePro-MoE 1e-2     | 31.6     | 65.4 | 50.6 | 73.0      | 65.6  | 43.3 |
> | GatePro-MoE 1e-3     | 31.5     | 65.3 | 50.7 | 73.1      | 65.4  | 43.1 |
> | GatePro-MoE 1e-4     | 31.5     | 65.2 | 50.7 | 73.0      | 65.5  | 43.2 |
> | MoE baseline | 30.5     | 64.8 | 49.8 | 72.8      | 64.6  | 42.0 |
>
> As shown above, using a larger λ (1e−1) does produce slightly higher improvements compared to 1e−4—for example a +0.4 gain on MMLU-Pro and +0.3 on MBPP. However, the improvements are relatively modest. In our paper, for 256 experts specifically, a larger λ indeed yields slightly better results, as your intuition suggests. We agree this is a meaningful ablation, and we will include this comparison and discussion in the revised version of the paper.
>
> Your review has greatly helped us strengthen the paper. We hope that our answers effectively resolve your concerns, and we would truly appreciate it if you could raise the score after reviewing our responses. Thank you again for your thoughtful review.

---

### Official Review · Reviewer_b2MX · 2025-10-30

**Soundness:** 2
**Presentation:** 3
**Contribution:** 2
**Rating:** 6
**Confidence:** 3

**Summary:**

The paper raises a method to cut down expert redundancy and enhance diversity by manipulating logits. It identifies the most similar expert pairs and introduces localized competition mechanisms, which suppress the logits of the loser. With experiments of decent scale, it reported consistent performance improvement and provided much analysis.

**Strengths:**

1. It reports better performance.
2. Large-scale experiments have been conducted.
3. Much analysis is provided.

**Weaknesses:**

1. Will such a small \(\lambda\) (1e-4) on the logits actually affect routing and training?

2. I don’t understand how you calculate the zero-token count. Is it calculated over the whole batch? It seems strange that so many experts could be unused after 1k steps. Did I miss something?

3. Could you also provide the loss curve and the training loss gain?

**Questions:**

1. I would like to raise the same questions as I raised in the 'Weakness' section.
2. How do you choose the size of lambda? Is there any ablation?

---

> ### Author Response · Authors · 2025-11-21
>
> Thank you very much for your thoughtful and constructive reviews. Each of your questions is helpful for improving the paper. We address them below.
>
> **Answer to Question 1**: Based on your question, we conducted additional ablation experiments using  λ = 1e-1, 1e-2, 1e-3, and 1e-4 on 256 experts and compared performance on MMLU-Pro, MMLU, BBH, HellaSwag, GSM8K, and MBPP with 256 experts:
>
> |  λ  | MMLU-Pro | MMLU | BBH  | HellaSwag | GSM8K | MBPP |
> | ---- | ---- | ---- | ---- | ---- | ---- | ---- |
> | GatePro-MoE 1e-1     | 31.9     | 65.3 | 50.9 | 73.1      | 65.6  | 43.5 |
> | GatePro-MoE 1e-2     | 31.6     | 65.4 | 50.6 | 73.0      | 65.6  | 43.3 |
> | GatePro-MoE 1e-3     | 31.5     | 65.3 | 50.7 | 73.1      | 65.4  | 43.1 |
> | GatePro-MoE 1e-4     | 31.5     | 65.2 | 50.7 | 73.0      | 65.5  | 43.2 |
> | MoE baseline | 30.5     | 64.8 | 49.8 | 72.8      | 64.6  | 42.0 |
>
> As shown above, using a larger λ (1e−1) does produce slightly higher improvements compared to 1e−4—for example a +0.4 gain on MMLU-Pro and +0.3 on MBPP. However, the improvements are relatively modest.
>
> We chose λ=1e−4 primarily for expert sparsity and stability considerations. When the total number of experts becomes very large (e.g., in future realistic industrial-scale MoE deployments), many experts will naturally receive extremely infrequent updates, especially in the early training stage. In such ultra-sparse regimes, using a very strong penalty such as λ=1e−1 could over-regularize the logits and destabilize the routing dynamics.
>
> In our paper, for 256 experts specifically, a larger λ indeed yields slightly better results, as your intuition suggests. We agree this is a meaningful ablation, and we will include this comparison and discussion in the revised version of the paper.
>
>
> **Answer to Question 2**: Yes, the zero-token count is computed over the whole batch at each training step.
> This behavior is closely related to model scale. For a relatively small model (e.g., around 1B parameters), experts tend to become active earlier during training, so the zero-token issue is less pronounced. In our experiments, however, the models are 7B and 13B. For these larger models, the MOE routing is more complex, and many experts only start to receive tokens much later.
> This delayed expert activation is exactly one of the challenges that GatePro is designed to address: during the early phase of training, GatePro rapidly promotes expert selection diversity, ensuring that all experts receive useful training signals much earlier than with standard MoE routing.
>
> **Answer to Question 3**: Certainly. Due to page limits, we did not include the baseline vs. GatePro loss curves in the revised version of the paper. We will add a loss-curve comparison figure in the revised paper. Below we show the training loss table of the 7B model extracted from wandb measured by the number of consumed tokens (in Billions):
>
> | Tokens (B) | 1B   | 25B  | 50B  | 75B  | 100B | 200B | 300B | 400B | 500B |
> | ---- | ---- | ---- | ---- | ---- | ---- | ---- | ---- | ---- | ---- |
> | MoE baseline   | 3.11 | 2.23 | 2.13 | 2.11 | 2.10 | 2.03 | 1.98 | 1.90 | 1.84 |
> | GatePro-MoE    | 3.05 | 2.19 | 2.10 | 2.07 | 2.06 | 2.01 | 1.95 | 1.88 | 1.83 |
>
> We observe that GatePro consistently achieves lower training loss than the baseline across all stages of training, with a more noticeable gap in the early and mid phases (e.g., 3.11 → 3.05 at 1B tokens, 2.13 → 2.10 at 50B tokens, 2.11 → 2.07 at 75B tokens). This aligns with the intended effect of improved early expert utilization and more stable routing dynamics.
>
> Your review is crucial to improving our paper. We hope that our answers effectively resolve your concerns, and we would truly appreciate it if you could raise the score after reviewing our responses. Thank you again for your thoughtful review.

---

### Official Review · Reviewer_BeJf · 2025-11-03

**Soundness:** 2
**Presentation:** 3
**Contribution:** 3
**Rating:** 4
**Confidence:** 4

**Summary:**

This paper proposes a novel method for Mixture-of-Experts (MoE) to balance the selection of experts. Overall, the paper is mostly clear in its writing and explanations and demonstrates an improvement over a naïve baseline. However, I was left with 2 main questions:

•	Is expert selection that is balanced actually a problem? I’m inclined to agree with the authors here that it makes sense to use all of the parameters possible in a model as much as possible, but lots of problems we encounter have a long-tail distribution. Due to this fact, does it make sense that activations of routings should also have a long-tail? Probably not, but is there anything that you can show (even somewhat anecdotally) with your proposed model that demonstrates why competitive propagation is more useful? I see that your accuracy improves in your tables and figures, but is there a clearer way to show why? For instance, looking at imbalanced classes in a test set?

•	Your experiments clearly show you beat the common MoE baseline (Fig 2, Table 1). Figures 3 & 4 are more ablations with the balancing. However, there are a lot of papers that have proposed improvements to Mixture-of-Experts and it would make sense to compare to some of those that have also improved over this same baseline. Could you show an experiment compared to another balancing router MoE method that has been published?

My second question about baselines comes from a desire to see the method compared to recent methods. Your related works section (2) has a lot of potential baselines that you could chose to consider. I’d like to see one of them.


Caption for Figure 1 should be more descriptive. The method is explained in the text, and in equations, and pseudo-code, but the figure is very opaque and it should describe to the reviewer what it they are looking at.

**Strengths:**

Interesting new method that beats a naive baseline.

**Weaknesses:**

Stronger baselines needed.

Caption for Figure 1 should be more descriptive. The method is explained in the text, and in equations, and pseudo-code, but the figure is very opaque and it should describe to the reviewer what it they are looking at.

**Questions:**

Why does the method matter? Re: my question about long-tail distributions above.

Please show another more recent method as a stronger baseline.

---

> ### Author Response · Authors · 2025-11-21
>
> We sincerely appreciate your thoughtful and constructive reviews. Our detailed responses follow.
>
> **Answer to Question 1**: Balanced expert selection is indeed a well-recognized problem in MoE training. Our analysis provides empirical evidence: comparing GatePro w/o balance vs. GatePro with balance across metrics like average cosine similarity, average angle, and spectral entropy clearly demonstrates that without balance mechanisms, expert utilization suffers significantly. Configurations without balance loss show totally different cosine similarity, angular separation, and entropy, indicating worse expert diversity and utilization.
>
> The question about long-tail distribution: In our paper, we focus on pretraining LLMs, where training data is specifically curated to be diverse and broadly representative across different domains, topics, and linguistic phenomena. If pretraining data exhibits severe long-tail distribution, this would itself be a data curation problem that undermines the goal of pretraining - to learn generalizable representations across diverse tasks. Our experimental setup reflects this principle: the 7B MoE model is trained on 500B tokens, while the 13B MoE model is trained on 1.2T tokens of diverse pretraining data. This extensive training ensures that models are sufficiently exposed to diverse domains, topics, and linguistic phenomena, making balanced expert utilization desirable for capturing this broad representational space. In such a setting, functional redundancy where similar experts are co-activated represents inefficient use of model capacity rather than a natural reflection of data distribution.
>
> However, investigating long-tail distributions would be valuable in post-training scenarios. For instance, during SFT or RLHF phases, future researchers could deliberately use long-tailed datasets to analyze how expert states evolve differently between MoE and GatePro-MoE when facing imbalanced data distributions. Such analysis would provide insights into how expert specialization adapts to different data regimes. However, this investigation falls outside the scope of our current work, which focuses on improving expert diversity during the pretraining phase where balanced, diverse data is the standard. We want to clarify that our work addresses different issue: functional redundancy and expert selection diversity rather than balance. Even with a perfectly balanced or naturally long-tailed activation distribution, if multiple experts learn redundant functions, this wastes model capacity. To demonstrate why competitive propagation helps beyond accuracy improvements:
>
> 1. Expert Utilization Analysis (Figure 3): Shows GatePro accelerates expert activation from earliest training stages, indicating more efficient capacity utilization.
>
> 2. Gating Similarity Analysis (Figure 4): Demonstrates lower cosine similarity and higher angular separation, proving experts develop more distinct functions.
>
>
>
> **Answer to Question 2**: In fact, we do include a comparison with a very recent MoE method in section 4.3: OLMoE, an open-source MoE model released recently that combines recent Top-k token-choice routing method, Switch-style load balancing loss (LBL method), and router z-loss for logit stabilization method - representing one of the most advanced MoE implementations in the research community.
>
> This is the reason why we dedicated Section 4.3 to evaluating GatePro on the open-source OLMoE-1B/7B architecture. Our goal was to demonstrate whether GatePro can provide additional benefits even on top of a model that already incorporates multiple recent MoE improvements. Table 2 clearly shows the performance comparison on OLMoE, with GatePro achieving consistent improvements across all evaluated benchmarks. This result is particularly significant because it demonstrates that GatePro's competitive propagation mechanism addresses a complementary aspect of MoE optimization. The consistent gains on top of OLMoE validate that GatePro can be integrated with state-of-the-art MoE methods to achieve further improvements.
>
> The caption of Figure 1: we thank the reviewer for this suggestion. We will revise the caption of Figure 1 to be more descriptive.
>
> Your review is crucial to improving our paper. We hope that our answers effectively resolve your concerns, and we would truly appreciate it if you could raise the score after reviewing our responses. Thank you again for your thoughtful review.

---

### Meta-Review · Area_Chair_cy7Z · 2026-01-07

**Summary:**

The submission proposes an expert pruning strategy based on cosine similarity and a localized competition mechanism, which in the end is summarized in Algorithm 1.  Effect size in improvement over baselines is small, but consistent.

**Reviewer Concerns:**

Reviewers raised concerns about the strength of baselines, (a lack of) theoretical justification of the cosine distance, and robustness to the lambda parameter.  The robustness to the lambda parameter appears to be demonstrated in additional experiments in the rebuttal.  The response to concerns about the strength of baselines is limited to that a specific library is used that has several options/variants. Given that the effect size is somewhat small, and that there are a very large number of published works on expert selection, this argument is less strong.

**Reviewer Scores:**

Reviewers were mixed with half indicating the submission is marginally above, and half indicating marginally below acceptance.  Given open concerns about the strength of baselines, and the relatively small effect size in the experiments, this appears to be a borderline reject case.

---

### Decision · Program_Chairs · 2026-01-26

Reject